# The Effect of Calcium Hydroxide, Triple Antibiotic Paste and Chlorhexidine on Pain in Teeth with Symptomatic Apical Periodontitis: A Randomised Controlled Trial

**DOI:** 10.3390/ijerph20043091

**Published:** 2023-02-10

**Authors:** Asma Munir Khan, Irfana Khursheed Ahmed Gangoo, Naila Amir Ali, Mansoor Khan, Muhammad Qasim Javed, Mustafa Hussein AlAttas, Ayman M. Abulhamael, Hammam Ahmed Bahammam, Loai Alsofi, Rayan Suliman Al Yahya

**Affiliations:** 1Department of Conservative Dental Sciences and Endodontics, College of Dentistry, Qassim University, Buraidah 52571, Saudi Arabia; 2Endodontics Specialty Clinics, Qassim University Medical City, Buraidah 52571, Saudi Arabia; 3Department of Operative Dentistry, Bolan University of Medical and Health Sciences, Quetta 87300, Pakistan; 4Department of Operative Dentistry, Foundation University College of Dentistry, Foundation University, Rawalpindi 44000, Pakistan; 5Department of Endodontics, Faculty of Dentistry, King Abdulaziz University, P.O. Box 80209, Jeddah 21589, Saudi Arabia; 6Department of Pediatric Dentistry, Faculty of Dentistry, King Abdulaziz University, Jeddah 21589, Saudi Arabia

**Keywords:** intracanal medicament, oral diseases, oral health, pain management, periapical periodontitis, calcium hydroxide, chlorhexidine, endodontics, randomized controlled trial

## Abstract

Background: One of the main reasons for post endodontic pain is the polymicrobial intracanal flora that may survive the initial disinfection. A single antimicrobial agent might not provide adequate disinfection, and an antimicrobial combination such as triple antibiotic paste was tested to achieve this goal. Aim: The study aimed to assess the efficacy of three intra-canal medicaments on post root canal preparation pain. Materials and Methods: Eighty patients with single-rooted necrotic teeth and symptomatic apical periodontitis were randomly assigned to four treatment groups (*n* = 20). Their preoperative pain was recorded on Wong-Baker’s FACES pain rating scale. After chemo-mechanical canal preparation, intracanal medications were applied to the groups (Group 1 (20% calcium hydroxide), Group 2 (2% chlorhexidine), Group 3 (tri-antibiotic paste), Group 4 (no medication (Control group)). Patients were instructed to record pain on the Wong-Baker FACES pain rating scale at 4, 48, 72 and 96 hrs, postoperatively. Pain scores were analyzed using a one-way ANOVA test and Tukey’s post hoc test and, if significant, Dunn’s test was used for pair-wise comparisons. The significance level was set at a *p*-value ≤ 0.05. Results: Tukey’s post hoc test revealed significantly lower pain scores in Group 3 compared with other groups at all follow up intervals. Dunnett’s test showed significantly lower pain in Group 3 compared with the Control group at 48, 72 and 96 hrs, postoperatively. Conclusions: Triple-antibiotic paste showed effective pain control as an intracanal medication on necrotic teeth with symptomatic apical periodontitis.

## 1. Introduction

Pain of endodontic origin is an undesirable event for both the patient and clinician. It has been estimated that 80% of patients who have pre-operative tooth ache continue to feel pain after endodontic treatment. Post-treatment pain is generally mild, and settles down within 72 hrs with analgesics such as NSAIDS or Acetaminophen [1]. However, sometimes patients continue to experience moderate to severe post-endodontic treatment pain for some days. The factors responsible for inter-appointment pain include pulpal remnants and chemical, microbial or mechanical irritation that is induced in the tissues of the peri-radicular area during endodontic treatment [2]. Moreover, pain of endodontic origin is frequently the result of bacterial by products, chronic inflammation, cytokine network activation and primed immune cells’ influx. Considering this, it might be a rational approach to minimize post-endodontic treatment pain through intra-canal placement of antimicrobial medications. Accordingly, to avoid the possible systemic side effects of orally administered antibiotics, local intra-canal antibiotic placement has been recommended over the last few decades [3].

An array of intra-canal medications is utilized in contemporary endodontic practice. One of these intra-canal medications is calcium hydroxide (CH). CH is employed as a multipurpose medicament in endodontic therapy, and is effective in alleviating pain by lowering the concentration of inflammatory exudates [4]. Chlorhexidine gluconate (CHX) is an antimicrobial medicament that targets Gram-negative and Gram-positive bacteria. CHX has a significant effect on resistant microorganisms residing in the root canal system, including *Enterococcus faecalis*, *Candida albicans* and anaerobic bacteria. Additionally, 0.1% to 2% CHX preparations are deemed toxicologically safe and effective for reducing pain during endodontic treatment [5]. Since endodontic infections are poly-microbial (a mixture of anaerobic/aerobic bacterial species), a single antimicrobial agent might not be adequate to achieve optimal intra-canal disinfection. Hence, a combination of three antibiotics comprising minocycline, metronidazole, and ciprofloxacin (triple-antibiotic paste [TAP]) has been recommended for intra-canal disinfection [6].

The present study aimed to evaluate the efficacy of three intra-canal medicaments on post root canal preparation pain in teeth with symptomatic apical periodontitis. This single-blind randomized control trial was designed to compare the effectiveness of different intra-canal medications on postoperative tooth pain, with a Control group. 

## 2. Materials and Methods

The study was conducted from March 2018 to April 2019 at the Department of Endodontics, Qassim University in Saudi Arabia after obtaining the approval from the institutional review board at the university (EA/6008/2018). This was a single-centered, single-blind randomized controlled clinical trial. The trial was registered with the Australia and New Zealand clinical trial registry (clinical trial number: ACTRN12622001513707). The participants were enrolled for the trial after being provided with complete information about the clinical trial and signing written informed consent. The primary outcome of the study was to measure the efficacy of three intra-canal medicaments on post canal preparation pain at 0, 4, 48, 72, 96 hrs, postoperatively. Efficacy of intra-canal medicament was defined as a reduction in the mean pain score values at time intervals of 0, 4, 48, 72, and 96 hrs, respectively, using Wong-Baker’s FACES pain rating scale (WFPRS). WFPRS is six-point scale comprising six faces (no hurt; hurts a little bit; hurts a little more; hurts even more; hurts a whole lot; hurts worst). The scale corresponds to 0, 2, 4, 6, 8, 10 on the numeric rating scale, and is used in combination with it (Figure 1) [7] During data analysis, for simplification, the values 2, 4, 6, 8, 10 were changed to 1, 2, 3, 4, and 5, respectively [8]. 

The sample size was calculated by utilizing PASS software version11 (Microsoft, Redmond, WA, USA). Two independent sample *t*-tests were applied, with a 95% confidence interval and 90% power. The minimum sample size needed was calculated as 80 (or 20 patients per group). The total sample was increased to 100 to make up for the anticipated drop-out of participants. Overall, 100 patients with age ranging between 16 and 50 years were enrolled in the study according to the following criteria.

### 2.1. Inclusion Criteria 

Patients were enrolled in the following cases:A single-rooted teeth (anterior and posterior);Teeth with symptomatic apical periodontitis; andTeeth with necrotic pulp that gave a negative response to vitality testing.

### 2.2. Exclusion Criteria 

Patients were excluded from the study in the following cases 

Non-restorable teeth;Presence of endo-perio lesions;Teeth associated with an acute or chronic apical abscess;Teeth with chronic periodontitis;Teeth with anatomical difficulties, such as open apices;Teeth with calcified canals;Teeth with severe dilacerations;Teeth with internal/external root resorption;Teeth with occlusal interferences;Patients who were allergic to medications that were used in the study;Patients who were taking medicines that could influence pain perception;Patients that had serious medical illness;Patients with systemic disorders; andPatients with immunocompromised diseases such as AIDS or HBV.

The participants were randomly assigned to four groups according to a simple random number table available online (www.random.org (accessed on 2 February 2018)) by an independent operator [9]. The teeth with symptomatic apical periodontitis that did not respond to electric pulp testing (Electric pulp tester; Sybron endo, Detroit, MI, USA) and a cold sensibility test (Roeko ENDO-FROST cold spray; Coltene, Alstatten, Switzerland) were diagnosed as necrotic (ICD 10 Code K 04.1) and were included in the study. The presence of acute apical inflammation (ICD 10 Code K 04.4) was assessed, based on patient history and response to percussion and palpation testing of the tooth under treatment. The acute periapical inflammation (ICD 10 Code K 04.4) was differentiated from other periapical conditions based upon the findings of the pre-operative periapical radiographs. Teeth that had acute pain on percussion and palpation but had minimal apical changes on the radiograph, such as periodontal ligament widening or minimal bone resorption in the apical area, were classified as having symptomatic apical periodontitis and were included in the study. Thereafter, the patients were given the information about WFPRS and asked to record their preoperative pain score on WFPRS.

All the cases were diagnosed and treated by a single endodontist with eight years of experience, under 3.5× magnification and using dental magnifying loupes (Univet Loupes, Rezzato, Italy). After confirming the eligibility, the offending tooth was anesthetized with 1.8 mL of 2% lidocaine, with 1:100,000 epinephrine (Octocaine 100, Novocol Pharmaceutical, Cambridge, ON, Canada) using the local infiltration (mandibular anterior/maxillary teeth) and inferior alveolar block (mandibular premolars). Rubber dam isolation was performed for each patient to prevent teeth contamination by saliva. 

The dental caries was removed followed by complete deroofing of pulp chamber. The size of the access cavities was not standardized and was dependent upon the size of the carious lesion. A diagnosis of pulp necrosis was confirmed at this step, and only those teeth showing no bleeding upon access preparation and a visibly necrotic, dark pulp tissue were included (ICD 10 Code K 04.1). The pre-endodontic tooth build-up was carried out with glass ionomer cement to ensure the proper isolation and inter-appointment sealing of the access cavity for cases in which access preparation extended to more than one surface. The glide path was prepared with stainless steel hand K-files, #06, #08, and #10 (Dentsply Maillefer, Switzerland). The working length was obtained by an electronic Apex locator (Dentaport ZX, J Morita, Kyoto, Japan) and verified by periapical radiograph. The root canal preparation was carried out to #20 (0.02 taper) with a hand K-file (Dentsply Maillefer, Switzerland). The Ni-Ti Rotary assorted files (Protaper Universal; Dentsply Maillefer, Switzerland) were then used to prepare the root canals using the crown down technique. Copious irrigation with 3% NaOCl was carried out. The canals were flooded with 17% EDTA solution (MD-Cleanser, Meta Biomed, Cheongju, Republic of Korea) for 3 min, post root canal preparation, to remove the inorganic component of the smear layer. 5 mL saline solution was used as the final rinse, and the canals were dried with sterile paper points (Dentsply Maillefer, Ballaigues, Switzerland). Intra-canal medicaments, using a Lentulo spiral, were placed according to the groups assigned to the patients in the random number table.

Group 1 received CH (20%) as intra-canal medicament; Group 2 received CHX (2%) gel; Group 3 received TAP (ciprofloxacin, minocycline, and metronidazole mixture in a proportion of 1:1:1 by weight); Group 4 received no dressing (Control group).

Excess medicaments coronal to the cementoenamel junction were removed with a moist cotton pellet, followed by placement of a sterile dry cotton pellet in the pulp chamber. Subsequently, the cavities were sealed with glass ionomer cement of minimum 3 mm thickness. The teeth were ground out of occlusion to prevent pain from occlusal forces. No medications were prescribed to the patients. Before dismissal, patients were instructed to record their pain intensity score on Wong-Baker’s FACES pain rating scale at 4, 48, 72, 96 hrs. Data were entered into an excel sheet and was later exported to SPSS. All analysis was carried out using SPSS version 24 (IBM, Armonk, NY, USA). The normal distribution of data was checked by creating histograms, and after confirmation, parametric tests were used. A one-way ANOVA test and Tukey’s post hoc test were utilized to perform multiple comparisons of pain scores between different groups at different time intervals. The level of significance was set at less than 0.05. Lastly, all the experimental groups were compared with the control group using Dunnett’s test.

## 3. Results

A total of 100 patients were initially evaluated for eligibility. Twelve patients were excluded from the study as they did not meet the inclusion criteria. However, eight of the patients refused to participate in the study. Hence, a total of eighty patients were included in the study (Figure 2). The patients were randomly divided into four groups according to the random number table. Group 1 was named Ca (OH)_2_, Group 2 as Chlorhexidine, Group 3 as Tri-antibiotic paste and Group 4 as the Control group. 

The mean age of the patients was 30 ± 2.34 years, and no statistically significant differences in baseline characteristics (age and gender) were observed between the groups (Table 1). Table 2 compares the effects of each medication on pain (WFPRS) scores between the four groups. The mean pain score and the standard deviation were recorded and tabulated for all four groups. The difference in the mean pain score between the groups was found be significant at 4 hrs, 48 hrs, 72 hrs and 96 hrs. Figure 3 and Table 2 compare the effects of each medication on pain (NRS) scores between the four groups.

The Tukey’s HSD test for multiple comparisons revealed that at 4 hrs there was a significant difference in pain scores of Group 3 and Group 2, and at 48 hrs and 72 hrs there was a significant difference in the pain score of Group 3 when compared with other groups. Lastly, at 96 hrs, there was significant difference in the pain scores of Group 3 in comparison with the Group 2 and Group 4. The Dunnett’s test showed a significant difference in the pain score of Control group (Group 4) and Tri-antibiotic group (Group 3) at 48 hrs, 72 hrs and 96 hrs (Table 3). Figure 4 summarizes the effects of the medications on postoperative pain over time. 

The Bartlett’s test was utilized to check the homogeneity of variance. It was found from the results that variance was homogenous for the preoperative, 48 hrs and 72 hrs pain scores. Conversely, non-homogenous variance was noted for the other time periods., i.e., 4 hrs and 96 hrs. Moreover, no adverse effects were reported by the patients in relation to medicaments that were used in this study.

## 4. Discussion

This study monitored the effects of different root canal medicaments in necrotic teeth with symptomatic apical periodontitis. Before the placement of the medicaments in the pulp canal, the smear layer was removed using an EDTA rinse to allow maximum penetration of the medicament in the dentinal tubules and the accessory canals. This allowed thorough disinfection of the canal space by effectively eliminating all the niches where bacteria may reside [10,11]. Only single-rooted teeth were included in order to eliminate the impact of varied and complex root canals systems encountered in multi-rooted teeth [12,13,14,15,16]. This study revealed information regarding the relative effectiveness of different intra-canal medicaments in alleviating post-operative pain. The most significant reduction of pain was seen in the TAP group. Although no significant difference in pain control was present at 4 hrs post op between the TAP and Control group, in the next four days, the pain scores in the TAP group were significantly lower than all the other groups. The microbial flora responsible for pulp space infection is polymicrobial, and the lower pain scores in the TAP group indicate the better antimicrobial activity of this medicament.

TAP is a combination of three different antibiotics, ciprofloxacin, metronidazole, and minocycline. Ciprofloxacin is a broad-spectrum antimicrobial agent which has bactericidal action against both Gram-negative and Gram-positive bacteria. Metronidazole is effective primarily against obligate anaerobic bacteria, while minocycline is a broad-spectrum tetracycline that mainly has a bacteriostatic action on bacteria through inhibition of bacterial protein synthesis. The synergistic effect of these three antibiotics in eliminating bacteria and achieving better post-operative pain control is in concurrence with the results of a previous study by Omaia et al. [17]. However, in that study, the TAP combination also had an anti-inflammatory drug in it, which might have helped in achieving good pain control due to its anti-inflammatory effect. Apart from this, tetracyclines can also inhibit matrix metalloproteinases, collagenase enzymes and fibroblast migration, which may lead to a decrease in bone/root resorption and early periapical healing [18,19]. Moreover, an anti-inflammatory mechanism of tetracyclines has also been explained [20].

In the current study, CH has the second lowest pain scores of all the groups. CH is the most widely used intra-canal medicament and has good antimicrobial activity. Due to the alkaline environment created in the apical area, CH also neutralizes periapical inflammation [21]. However, recent studies have revealed a mechanism that causes a decrease in antimicrobial activity of CH due to buffering by dentin and a decreased penetration of CH in the dentinal tubules [22]. Certain bacteria have been shown to be resistant to the antimicrobial effects of CH [23]. The lower pain control exhibited by CH compared to TAP in the present study is in concurrence with the findings of Prasad et al. [24]. However, in the aforementioned study, patients with acute irreversible pulpitis and chronic irreversible pulpitis with acute apical periodontitis were also included; these are both conditions in which the pulp space does not contain a significant number of bacteria, and the pain is mainly due to acute inflammatory response. This indicates an as-yet unknown ability of TAP to achieve pain control via modulation of the host inflammatory response.

CHX is a broad-spectrum antimicrobial effective against Gram-positive and Gram-negative bacteria [25]. CHX is also effective against *E. faecalis* [26] and *Candida albicans* [27]. However, multiple studies have shown the inactivation of CHX upon contact with dentine matrix [22,28]. Singh et al. reported lowest pain scores post operatively when using a combination of CHX-CH as intra-canal medicament in necrotic teeth with symptomatic apical periodontitis [29]. This demonstrates that the addition of CH to CHX improved the ability of the medicament to modulate the host inflammatory response. In the present study, the presence of highest pain score post operatively was observed in the CHX, which might be due to decreased antimicrobial activity due to inhibition or due to an inability to affect the host inflammatory response.

Presence of minocycline has been reported to cause discoloration of teeth when used as an intra-canal medicament. This may be avoided by using dentine bonding agent in the pulp chamber to prevent minocycline uptake by coronal dentine. Another method may be alternating minocycline with cefaclor to make modified TAP, which has been reported to have good antimicrobial activity [30]. The fear of promoting antibacterial resistance by using antibiotics is another factor preventing the use of TAP as an intra-canal medicament. However, in the authors’ opinion, the use of antibiotic prodrugs requiring bacteria specific enzymes to release the active drug [31] and appropriate case selection to restrict the use of TAP in selective cases with necrotic pulps can alleviate these concerns. Yassen et al. have also shown a decrease in dentine micro hardness with long-term use of TAP [32]. However, in that study, the medicament was placed in the canal for more than a month, whereas in most clinical situations, the medicaments are not in place for more than 2 weeks. GIC was used to temporize all the access preparations, while ensuring at least 3 mm thickness of the filling material to prevent leakage of bacteria. GIC has good mechanical properties and bonds with the tooth structure as well, preventing any contamination of the root canals due to restoration dislodgement or leakage [33].

The current study has certain limitations. Firstly, in the present study, cotton pellets were used as endodontic spacers and to wipe off excess medicaments from the access cavity walls. This step might leave residual cotton fibres on the access cavity walls, and can interrupt the seal of the temporary restoration as well as acting as a potential substrate for surviving bacteria [34]. Secondly, the procedures were performed without the aid of an endodontic microscope. Microscopes have been shown to improve endodontic outcomes by helping in the detection and removal of necrotic debris and the management of canal variations and calcifications [35]. Lastly, the size of the access cavities could not be standardized in the present study due to the varying size of the carious lesions in the different teeth that were enrolled in the study. Study models utilizing conservative access preparation may be used in future studies to eliminate this factor.

## 5. Conclusions

Triple-antibiotic paste showed effective pain control as an intra-canal medicament. Under the conditions of the present study, it is recommended to use TAP in necrotic teeth as an intra-canal medicament to achieve effective post-operative pain control. However, caution must be exercised in use of TAP as a first-choice treatment due to the potential of introducing antibiotic resistance, at least until a bacteria-specific antibiotic delivery system is developed. Further investigation is also required to find out the exact mechanism through which TAP affects the host inflammatory response, and find out what the impact of TAP in controlling postoperative pain in cases of irreversible pulpitis would be.

## Figures and Tables

**Figure 1 ijerph-20-03091-f001:**
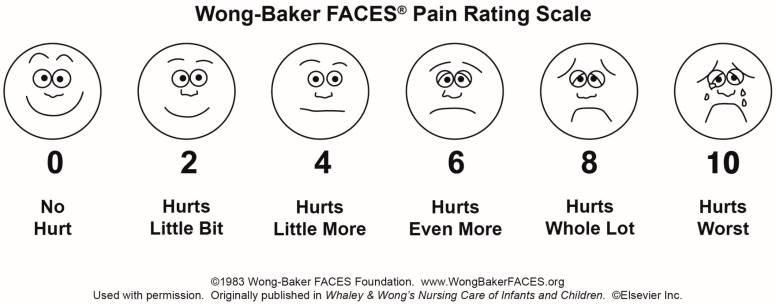
Wong-Baker’s FACES pain rating scale (used with permission).

**Figure 2 ijerph-20-03091-f002:**
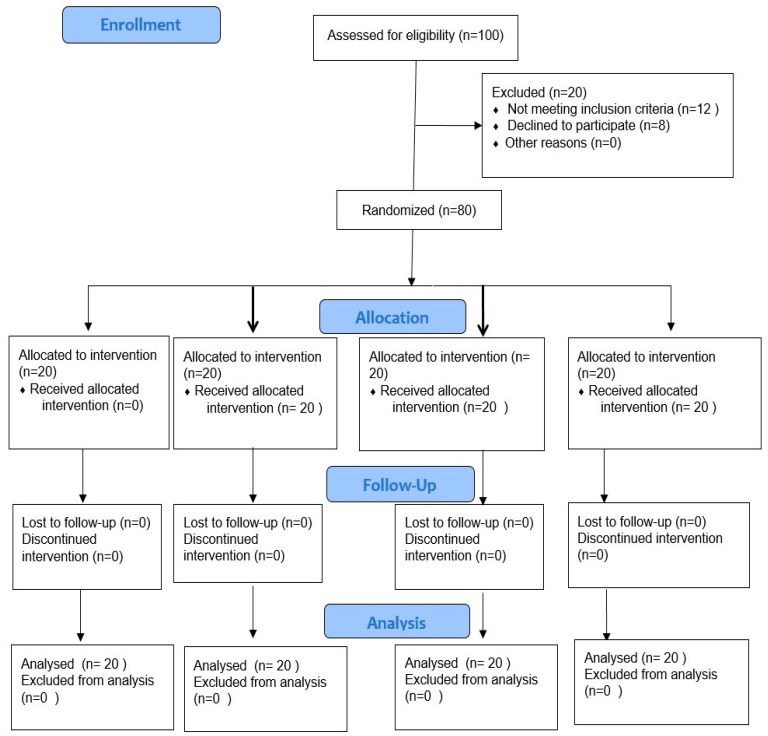
CONSORT flow diagram for the study.

**Figure 3 ijerph-20-03091-f003:**
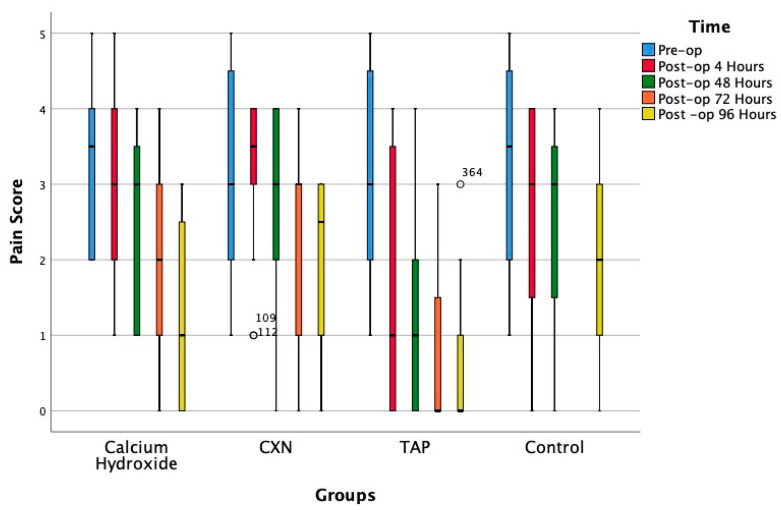
Box plot showing median (and range) pain scores in the four groups.

**Figure 4 ijerph-20-03091-f004:**
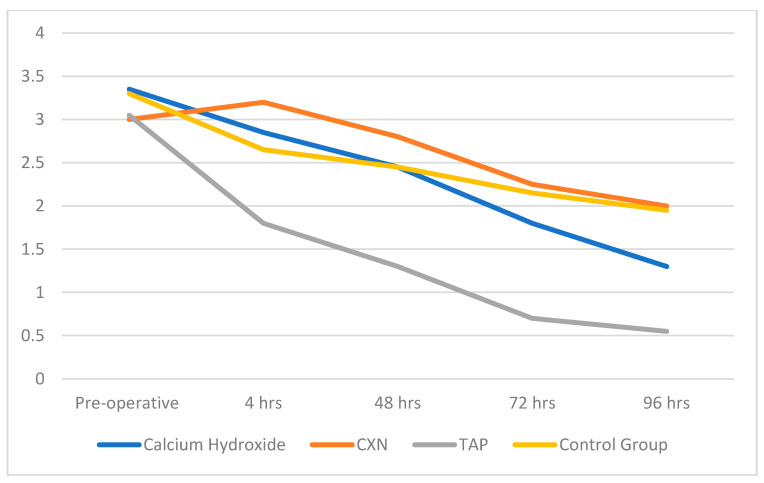
Line chart showing mean pain scores at different time intervals by treatment group.

**Table 1 ijerph-20-03091-t001:** Descriptive statistics of study sample.

	Calcium Hydroxide	Chlorohexidine	Tri-Antibiotic	Control	*p*-Value
Age mean (SD) years	30.4 (2.23)	29.8 (2.27)	30.1 (2.45)	29.7 (2.41)	*p* > 0.05
Male	10 (50%)	9 (45%)	8 (40%)	11 (55%)	
Female	10(50%)	11 (55%)	12 (60%)	9(45%)	

Significance was set at *p* ≤ 0.05.

**Table 2 ijerph-20-03091-t002:** Median (range) and mean (± SD) pain (WFPRS) scores in the four groups.

Intra-Canal Medicaments							
Time Period		Group 1 Calcium Hydroxide (*n* = 20)	Group 2 Chlorhexidine (*n* = 20)	Group 3 Tri-Antibiotic (*n* = 20)	Group 4 Control Group (*n* = 20)	* *p*-Value	f-Value
Preoperative	Median (range)	3.5 (2–5)	3 (1–5)	3 (1–5)	3.5 (1–5)	0.816	0.313
	Mean (SD)	3.35 (1.18)	3.00 (1.52)	3.05 (1.46)	3.30 (1.41)		
4 hrs	Median (range)	3 (1–5)	3.5 (1–4)	1 (0–4)	3 (0–4)	* 0.010	4.032
	Mean (SD)	2.85 (1.26)	3.20 (1.00)	1.80 (1.64)	2.65 (1.30)		
48 hrs	Median (range)	3 (0–4)	3 (0–4)	1 (0–4)	3 (0–4)	* 0.003	5.196
	Mean (SD)	2.45 (1.23)	2.80 (1.28)	1.30 (1.21)	2.45 (1.39)		
72 hrs	Median (range)	2 (0–4)	3 (0–4)	0 (0–3)	2 (0–4)	* 0.000	7.115
	Mean (SD)	1.80 (1.24)	2.25 (1.20)	0.70 (0.97)	2.15 (1.30)		
96 hrs	Median (range)	1 (0–3)	2.5 (0–3)	0 (0–3)	2 (0–4)	* 0.000	7.010
	Mean (SD)	1.30 (1.21)	2.00 (1.12)	0.55 (0.82)	1.95 (1.35)		

* Tukey’s HSD test; significant was set at *p* ≤ 0.05.

**Table 3 ijerph-20-03091-t003:** Comparison of experimental groups versus Control group (Group 4) using Dunnett’s Test.

Time Period	Experimental versus Control Group	*p*-Value
4 hrs	1 vs. 4	0.934
	2 vs. 4	0.419
	3 vs. 4	0.115
48 hrs	1 vs. 4	1.000
	2 vs. 4	0.722
	3 vs. 4	* 0.016
72 hrs	1 vs. 4	0.675
	2 vs. 4	0.987
	3 vs. 4	* 0.001
96 hrs	1 vs. 4	0.186
	2 vs. 4	0.998
	3 vs. 4	* 0.001

*: Significant at *p* ≤ 0.05.

## Data Availability

As per the instructions of ethical committee, the data will be kept in a password-protected computer in the department. Only the principal investigator and the statistician have access to the data. Considering this, the patient data cannot be shared.

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
