# Peer review of "The Effect of Calcium Hydroxide, Triple Antibiotic Paste and Chlorhexidine on Pain in Teeth with Symptomatic Apical Periodontitis: A Randomised Controlled Trial"

_ijerph, 2023, doi:10.3390/ijerph20043091_

Round 1

Reviewer 1 Report

Dear Authors,

Thank You for a pleasure to read Your work.

I have several comments and notes to improve Your work.

First, please, check English; there are some mistakes.

Abstract

Journal allows to write non-structural abstract but for RCT it is better to make it structural (according usual ‘Background- Aim- Materials and Methods-Results-Conclusion’)

You have no statistical signature in abstract, please, add

Please, check key words in MeSH.

Materials and Methods

Please, mark inclusion/exclusion criteria as list or table, not leave them as simple text.

For diagnosis also add ICD-10 code.

Statistics

Please, write distribution normality test. Also, it is better for dynamics assessment and 2 factors (period and group) to use in Your case two-way ANOVA test. Please, add it if it is possible.

Please, add the table 1 with descriptive statistics with groups characteristics before the intervention.

Please, add for ‘Table 1. Postoperative pain mean values of experimental and control groups’ medians, min-max and confidence interval.

  For table 2? Please? mark the significant measures.

Discussion

Please, use probability for your measures.

Please, write limitations for your study

Also, check structure of your article according to CONSORT checklist because You missed several sections and try to make your article according it.

Sincerely, Reviewer

Author Response

Dear Reviewer,

Thanks for reviewing our manuscript. You comments have have enabled us to make significant improvement in the manuscript.

Please find attached our response.

Reviewer 2 Report

A very interesting article on an important topic in endodontics, namely postoperative pain after drug placement. The authors studied patients with root canal-treated teeth for 96 hours, after careful selection of inclusion and exclusion criteria for this study.

The presentation of the data is adequately done and clear.

A few asceptions should definitely be explained and discussed in more detail:

First: 
Teeth were treated endodontically, however, no information is currently available in the paper as to which practitioners performed this. This information needs to be supplemented, especially with regard to the standardisation of the treatment process and the resulting statement. It is of great interest to the reader to know whether the treatments were carried out by endodontists or dentists. What experience did the dentists have? Furthermore, the question of whether treatment was carried out with magnifying glasses or an operating microscope.

Secondly:
Cotton wool pellets were used for cleaning excess of the medicated filling as well as a cotton wool pellet was also placed in the cavity before placing the temporary filling. It is very important to discuss here that these cotton pellets had shown in studies that they leave fibres behind that lead to leakage of fillings. This aspect must also be discussed intensively, because its consequences are elementary for possible recontamination. Indeed, it is conceivable that even possibly the TAP successfully leading to freedom from pain in this study might have been more effective in preventing bacteria of a coronal leak than the other medicinal inserts.

Thirdly:
How extensive were the access cavities? Again, differences in cavity size may potentially have an impact on the provisional filling with glass ionomer cement. Please add information.

Fourth:
Regarding the possible formation of resistance, this has already been discussed. Here it would be important to know what significance the authors see in freedom from pain and whether TAP would make sense here as a first-choice medication or whether resistance would not have to be feared here if TAP were used as a first-choice medication.

Author Response

(The authors gave the same response as above.)

Reviewer 3 Report

My comments and suggestions for authors are written in the attached file.

Author Response

(The authors gave the same response as above.)

Round 2

Reviewer 1 Report

Dear Authors,

Thank You for a pleasure to read You work.

I have small recommendation to improve Your work.

For figure 3, please, change graph for box plots, it will be more appropriate.

Also, it is not exactly clear that You used one-way ANOVA for time or group comparison.

Sincerely, Reviewer

Author Response

Dear Reviewer, 

Thanks for reviewing our manuscript. Please find attached the response letter. 

Regards

Reviewer 3 Report

I consider the study can be published because they made the requested modifications.

Author Response

Dear Reviewer, 

Thanks for reviewing and recommending our manuscript for acceptance.

Kind Regards, 

Muhammad Qasim Javed
